# Factors associated with receipt of adequate antenatal care among women in Rwanda: A secondary analysis of the 2019–20 Rwanda Demographic and Health Survey

Olive Tengera[1]*, Laetitia Nyirazinyoye[1], Pamela Meharry[1,2], Reverien Rutayisire[1], Stephen Rulisa[1], Zelalem T. Haile[3]

1 College of Medicine and Health Sciences, University of Rwanda, Kigali, Rwanda, 2 Department of Human Development Nursing Science, University of Illinois Chicago, Chicago, Illinois, United States of America, 3 Department of Social Medicine, Ohio University Heritage College of Osteopathic Medicine, Dublin, Ohio, United States of America

* tengera.olive@gmail.com

**Data Availability Statement:** Because this is a secondary data analysis of the RDHS, data are owned by a third-party organization it is not

## Abstract

### Background

Every year, antenatal care (ANC) remains a life-saving health intervention for millions of pregnant women worldwide. Yet, many pregnant women do not receive adequate ANC, particularly in sub-Saharan Africa. The study aimed to determine the factors associated with the receipt of adequate ANC among pregnant women in Rwanda.

### Methods

A cross-sectional study was conducted using the 2019–2020 Rwanda Demographic and Health Survey data. The study included women aged 15–49 years who had a live birth in the previous five years (n = 6,309). Descriptive statistics and multivariable logistic regression analyses were performed.

### Results

Overall, 27.6% of participants received adequate ANC. The odds of receiving adequate ANC were higher among those in the middle household wealth index (AOR 1.24; 1.04, 1.48) and rich index (AOR 1.37; 1.16, 1.61) compared to those in the poor wealth index category. Similarly, having health insurance was positively associated with receiving adequate ANC (AOR 1.33; 1.10, 1.60). The odds of receiving adequate ANC were lower among urban dwellers compared to rural (AOR 0.74; 0.61, 0.91); for women who wanted pregnancy later (AOR 0.60; 0.52, 0.69) or never wanted pregnancy (AOR 0.67; 0.55, 0.82) compared to those who wanted pregnancy; for women who perceived distance to a health facility as a big problem (AOR 0.82; 0.70, 0.96) compared to those that did not; and for women whose ANC was provided by nurses and midwives (AOR 0.63; 0.47, 0.8), or auxiliary midwives (AOR 0.19; 0.04, 0.82) compared to those who received ANC from doctors.

permissible to share with the public. However, the data is publicly available with permission from the MEASURE DHS Data Archive at ICF International and obtained upon request via the following link www.measuredhsprogram.com. The authors confirm they did not have any special access privileges to the data.

**Funding:** The author(s) received no specific funding for this work.

**Competing interests:** The authors have declared that no competing interests exist.

## Conclusion

The prevalence of women who receive adequate ANC remains low in Rwanda. Effective interventions to increase access and utilization of adequate ANC are urgently needed to further improve the country's maternal and child health outcomes.

## Introduction

Maternal and neonatal mortality are two of the most significant inequities in healthcare today. In 2017, 86% of global maternal deaths occurred in Sub-Saharan Africa (SSA) and Asia, with the former region accounting for two-thirds of mortality [1]. Similarly, SSA has the highest neonatal mortality rate, accounting for 43% of global newborn deaths [2]. Maternal and neonatal outcomes could be significantly improved with adequate antenatal care (ANC) by skilled providers throughout pregnancy [3–5].

Adequate ANC is defined based on the timing of the visit, frequency of visits, and receipt of components of care provided at each ANC contact [6]. The World Health Organization (WHO) 2016 guidelines recommend that all pregnant women receive early and regular quality care, which includes eight ANC contacts, double the number of the previously recommended four ANC-focused visits [6]. The first contact should be in the first trimester of pregnancy (at 12 weeks), and subsequently, two contacts in the second trimester (at 20 and 26 weeks) and five contacts in the third trimester (at 30, 34, 36, 38, and 40 weeks) [7]. The goal of ANC is to identify risk factors and chronic threats, establish baseline health status, and monitor and evaluate women and their fetuses regularly to prevent and avoid complications during pregnancy [8]. During ANC provision, pregnant women learn about healthy behaviors, disease prevention, and social, emotional, and psychological support from skilled health care providers (HCP) at a critical time of their life [6].

A review of Demographic and Health Surveys (DHS) and Multiple Indicator Cluster Surveys (MICS) conducted in 54 low- and middle-income countries (LMIC) [9] reported that women in the Central and Southern Asia regions had the highest prevalence of timely initiation of ANC, and women in the Caribbean and Latin America were most likely to reach eight or more ANC contacts. The review also reported that 44.3% initiated ANC in the first trimester, 49.9% had one or more ANC visits, and only 11.3% had eight visits [9]. Women that received more ANC content had about five times higher odds of attending four or eight ANC contacts ($P<0.001$) [9]. Women less likely to achieve adequate ANC included poor, single, lower-educated, rural dwellers, larger households, with lower birth intervals, higher parity, and non-health facility birth [9]. A South African study reported that lack of information, poor quality or insufficient health services, and cultural beliefs and practices also affected adequate ANC [10].

Access to adequate ANC is essential for HCP to intervene with potentially life-saving management. Most deaths are preventable if detected and treated early by a skilled HCP, which may reduce maternal morbidity and mortality by 20% [4]. Another consideration is the live-born infant with a deceased mother; a Nigerian survey showed that only about a third of mother-less infants survived beyond five years of age [11]; therefore, maternal survival is vital for child survival.

In Rwanda, ANC coverage with a skilled HCP has increased to about 99% of women [12], though ANC utilization (attending at least four visits) was only 47.6% [5], and distance to health facilities remains a barrier [13]. The WHO introduced the eight-contacts model in

2016, and Rwanda is planning to implement this model soon [3, 12]. This paper aimed to determine the factors associated with receiving adequate ANC services among pregnant women in Rwanda. We hypothesized that the receipt of adequate ANC would be independently associated with sociodemographic and access to care-related factors.

## Materials and methods

### Study design and setting

This is a secondary data analysis of the 2019–2020 Rwanda Demographic and Health Survey (RDHS). The 2019–20 RDHS is a cross-sectional survey [5] conducted in collaboration with the National Institute of Statistics of Rwanda (NISR), the Ministry of Health, and the DHS Program of ICF International.

In brief, a two-stage stratified random sampling method was used to select participants. The first stage involved selecting sample points (clusters) or enumeration areas (EAs) delineated for the 2012 Rwanda Population and Housing Census (RPHC), which involved 500 selected clusters, 112 in urban areas and 388 in rural areas. There were 14,675 women aged 15–49 years eligible for the individual interview in selected households. Nearly all women were interviewed for the RDHS (n = 14,634), resulting in a response rate of 99.7 percent. Detailed information about sampling and data collection methodology is available elsewhere [5].

The sampling frame for the current study included women aged 15–49 years who had a live birth within the five years preceding the 2019–20 RDHS (n = 6,302) data collection. We excluded women with missing data on ANC visits for the most recent live birth (n = 142) and other variables in the current study (n = 121). The final sample for the analysis comprised 6,039 women.

### Study variables

The outcome of interest was the receipt of adequate ANC, which was constructed by first creating an index based on responses to three questions 1) the total number of ANC visits, 2) the timing of the first ANC visit, and 3) components of ANC received. In the 2019–20 RDHS. Women who reported having a live birth in the past five years were asked to report on the total number of ANC visits, the number of months pregnant at the time of the first ANC visit, and components of ANC received during their ANC visits. The number of visits was coded as none to three, and at least four. The Rwanda Ministry of Health (MoH) ANC guidelines [12] state that the timing of the first visit should be within the first three months of pregnancy. This variable was coded as yes or no and defined as the first ANC visit within the first three months of pregnancy. Components of ANC were defined based on four ANC components reported in the 2019–20 RDHS [5] and included urine sample test, blood sample test, blood pressure measurement, and mid-upper arm circumference measurement. Then a dichotomous variable for receipt of adequate ANC was created and coded as 1 (yes) and 0 (no) if respondents had received at least 4 ANC visits, had the first visit within three months of pregnancy, and had received all four components of ANC. It was coded as 1 and 0 otherwise.

Based on the RDHS final report and existing literature, the following variables were selected: age (15–24, 25–34, 35–44, 45 years and above), marital status (never married, married/living together, and divorced/separated/widowed), education (none, primary, and secondary or higher), household wealth index (poor, middle, rich), employment (no, yes), health insurance (no, yes), residence (rural, urban), pregnancy intention (wanted then, wanted later, wanted no more), distance to nearest health care facility (big problem, not a big problem), birth order (1st, 2nd, 3rd, 4th child and above) and ANC provider (doctor, nurse or midwife, auxiliary midwife).

### Ethics approval

This study was a secondary analysis of RDHS 2019–20 which was registered and approved by the University of Rwanda College of Medicine and Health Sciences Institutional Review Board (CMHS/IRB/212/2021). The authors received a permission letter to access the database from the Measure DHS archive at ICF Institutional Review Board. Verbal informed consent was obtained before the interviews in the original population-based survey and reported in the aggregate format. Data are available to the general public by request in different formats from the Measure DHS website [www.measuredhsprogram.com].

### Statistical analysis

Categorical variables were presented as frequencies and proportions. The Rao-Scott chi-square test was used to compare differences in the receipt of adequate ANC by each factor. The factors included age, marital status, education, household wealth index, health insurance, distance, place of residence, employment, pregnancy intention, birth order, and type of ANC provider. All factors were entered simultaneously into the regression model regardless of statistical significance in the bivariate analysis. A multivariable-adjusted logistic regression was used to determine the independent association between each factor and receipt of adequate ANC. Odds ratio (OR) and 95% confidence interval (CI) were presented. A $p < 0.05$ was considered statistically significant. Complex sample design elements were applied for all analyses, including stratification, clustering, and sample weights. All analyses were performed using SAS OnDemand for Academics (SAS Institute Inc., Cary, NC).

## Results

Table 1 presents the characteristics of the study sample. The majority, 3949 (48.8), were aged between 25 and 34 years. The majority, 4896 (81.2%), were married, and 3928 (64.6%) had a primary education level. The poor constituted the highest proportion of the household wealth index, 2583 (41.7%), and almost a quarter, 1463 (24.4%), were unemployed, and 1066 (18%) did not have health insurance. Over half of the women, 3550 (58.4%), intended to get pregnant, and 2750 (34.0%) had a birth order of the 4th child or above. The majority, 4739 (82.2%), lived in a rural area, and 4652 (76.8%) perceived the distance to a health facility as a big problem. Only 1691 (27.6%) received adequate ANC; most providers were nurses or midwives, 5727 (95.0%).

 Table 2 shows the results of the Rao-Scott chi-square test used to compare differences in the receipt of adequate ANC by each factor. Adequacy of ANC received significantly differed by the women's age ($X^2 = 9.54$, df = 3, p = 0.023), marital status ($X^2 = 10.48$, df = 2, p = 0.005), education ($X^2 = 27.21$, df = 2, p < 0.001), household wealth index ($X^2 = 36.81$, df = 2, p < .001), health insurance ($X^2 = 23.09$, df = 1, p < 0.001), pregnancy intention ($X^2 = 85.88$, df = 2, p < 0.001), birth order ($X^2 = 22.32$, df = 3, p < 0.001), distance to a health facility ($X^2 = 13.62$, df = 1, p = 0.002) and the presence of a skilled provider at the ANC visit ($X^2 = 26.18$, df = 2, p < 0.001).

 Table 3 shows the crude odds ratio (COR) and multivariable-adjusted odds ratio (AOR) between the women's characteristics and receipt of adequate ANC. In the AOR, household wealth index, health insurance, residence, pregnancy intention, distance to the health facility, and ANC provider were significantly associated with the receipt of adequate ANC. Women from a middle and rich wealth indexes had higher odds of receiving adequate ANC (AOR = 1.24: CI = 1.04–1.48; p = 0.018) and (AOR = 1.37: CI = 1.16–1.61; p<0.001) respectively, compared to women from poor household wealth index. Compared to those without health insurance, women with insurance were more likely to receive adequate ANC (AOR

**Table 1. Demographic characteristics of the respondents (n = 6039).**

|  | Overall n (wt. %) |
|---|---|
| **Age group (years)** |  |
| 15–24 | 1247 (15.9) |
| 25–34 | 3949 (48.8) |
| 35–44 | 2652 (32.8) |
| 45 and above | 204 (2.5) |
| **Marital status** |  |
| Never married | 636 (10.1) |
| Married/living together | 4896 (81.2) |
| Divorced/separated/widowed | 507 (8.7) |
| **Education** |  |
| No education | 637 (10.7) |
| Primary | 3928 (64.6) |
| Secondary and higher | 1474 (24.7) |
| **Household wealth index** |  |
| Poor | 2583 (41.7) |
| Middle | 1151 (19.5) |
| Rich | 2305 (38.8) |
| **Employment** |  |
| No | 1463 (24.4) |
| Yes | 4576 (75.6) |
| **Health insurance** |  |
| No | 1066 (18.0) |
| Yes | 4973 (82.0) |
| **Place of residence** |  |
| Rural | 4739 (82.2) |
| Urban | 1300 (17.8) |
| **Intention to pregnancy** |  |
| Wanted then | 3550 (58.4) |
| Wanted later | 1692 (28.2) |
| No more | 797 (13.3) |
| **Birth order** |  |
| 1st | 2049 (25.3) |
| 2nd | 1821 (22.5) |
| 3rd | 1472 (18.2) |
| 4th and above | 2750 (34.0) |
| **Distance to health facility** |  |
| Big problem | 4652 (76.8) |
| Not a Big problem | 1387 (23.2) |
| **Adequacy of ANC received** |  |
| Inadequate | 4348 (72.4) |
| Adequate | 1691 (27.6) |
| **ANC provider** |  |
| Doctor | 291 (4.6) |
| Nurse or midwife | 5727 (95.0) |
| Auxiliary nurse | 21 (0.3) |

Abbreviations Wt.%: Weighted percent

**Table 2. Characteristics of the study sample by adequacy of ANC received (n = 6039).**

| | | Adequacy of ANC received | | $X^2$ (df) | P |
| --- | --- | --- | --- | --- | --- |
| | | Inadequate | Adequate | | |
| | | n (Wt. %) | n (Wt. %) | | |
| **Age group (years)** | | | | 9.54 (3) | **0.023** |
| | 15–24 | 772 (73.8) | 273 (26.2) | | |
| | 25–34 | 2002 (70.5) | 866 (29.5) | | |
| | 35–44 | 1447 (74.1) | 512 (25.9) | | |
| | 45 and above | 126 (76.4) | 40 (23.6) | | |
| **Marital status** | | | | 10.48 (2) | **0.005** |
| | Never married | 487 (77.3) | 149 (22.7) | | |
| | Married/living together | 3479 (71.4) | 1417 (28.6) | | |
| | Divorced/separated/widowed | 382 (75.5) | 125 (24.5) | | |
| **Education** | | | | 27.21 (2) | **< .001** |
| | No education | 489 (76.1) | 148 (23.9) | | |
| | Primary | 2888 (74.1) | 1040 (25.9) | | |
| | Secondary and higher | 971 (66.3) | 503 (33.7) | | |
| **Household wealth index** | | | | 36.81 (2) | **< .001** |
| | Poor | 1985 (76.8) | 598 (23.2) | | |
| | Middle | 823 (71.7) | 328 (28.3) | | |
| | Rich | 1540 (68.0) | 765 (32.0) | | |
| **Employed** | | | | 0.03 (1) | 0.867 |
| | No | 1043 (72.2) | 420 (27.8) | | |
| | Yes | 3305 (72.4) | 1271 (27.6) | | |
| **Health insurance** | | | | 23.09 (1) | **< .001** |
| | No | 844 (79.3) | 222 (20.7) | | |
| | Yes | 3504 (70.9) | 1469 (29.1) | | |
| **Place of residence** | | | | 0.28 (1) | 0.599 |
| | Rural | 3445 (72.6) | 1294 (27.4) | | |
| | Urban | 903 (71.5) | 397 (28.5) | | |
| **Pregnancy intention** | | | | 85.88 (2) | **< .001** |
| | Wanted then | 2440 (67.8) | 1150 (32.2) | | |
| | Wanted later | 1311 (78.3) | 381 (21.7) | | |
| | No more | 637 (79.9) | 160 (20.1) | | |
| **Birth order** | | | | 22.32 (3) | **< .001** |
| | 1st | 1031 (71.2) | 426 (28.8) | | |
| | 2nd | 952 (69.4) | 434 (30.6) | | |
| | 3rd | 792 (70.1) | 337 (29.9) | | |
| | 4th and above | 1572 (76.4) | 494 (23.6) | | |
| **Distance to health facility** | | | | 13.62 (1) | **0.002** |
| | Big problem | 3290 (71.1) | 1362 (28.9) | | |
| | Not a big problem | 1058 (76.7) | 329 (23.3) | | |
| **ANC provider** | | | | 26.18 (2) | **< .001** |
| | Doctor | 177 (57.7) | 114 (42.3) | | |
| | Nurse or midwife | 4153 (73.0) | 1574 (27.0) | | |
| | Auxiliary nurse | 18 (88.6) | 3 (11.4) | | |

Abbreviations Wt.%: Weighted percent; $X^2$: Rao Scott chi-square; df: degrees of freedom

**Table 3. Factors associated with receipt of adequate ANC (n = 6039).**

| Characteristics | | | Unadjusted | | Adjusted | |
|---|---|---|---|---|---|---|
| | | | OR (95% CI) | *P* | OR (95%CI) | *P* |
| **Age group (years)** | | | | | | |
| | 15–24 | | Reference | | Reference | |
| | 25–34 | | 1.18 (1.02, 1.37) | 0.025 | 1.06 (0.90, 1.26) | 0.470 |
| | 35–44 | | 0.99 (0.82, 1.19) | 0.903 | 1.00 (0.79, 1.27) | 0.998 |
| | 45 and above | | 0.87 (0.58, 1.33) | 0.528 | 0.97 (0.61, 1.54) | 0.890 |
| **Marital status** | | | | | | |
| | Never married | | Reference | | Reference | |
| | Married/living together | | 1.37 (1.12, 1.67) | **0.003** | 1.07 (0.84, 1.36) | 0.584 |
| | Divorced/separated/widow | | 1.11 (0.81, 1.51) | 0.522 | 1.00 (0.71, 1.40) | 0.978 |
| **Education** | | | | | | |
| | No education | | Reference | | Reference | |
| | Primary | | 1.12 (0.91, 1.38) | 0.304 | 0.97 (0.79, 1.19) | 0.801 |
| | Secondary and higher | | 1.63 (1.27, 2.09) | **0.002** | 1.21 (0.93, 1.59) | 0.147 |
| **Household wealth index** | | | | | | |
| | Poor | | Reference | | Reference | |
| | Middle | | 1.31 (1.10, 1.56) | **0.003** | 1.24 (1.04, 1.48) | **0.018** |
| | Rich | | 1.56 (1.34, 1.81) | **< .001** | 1.37 (1.16, 1.61) | **< .001** |
| **Employed** | | | | | | |
| | No | | Reference | | Reference | |
| | Yes | | 0.99 (0.84, 1.16) | 0.866 | 1.01 (0.86, 1.18) | 0.955 |
| **Health insurance** | | | | | | |
| | No | | Reference | | Reference | |
| | Yes | | 1.57 (1.30, 1.89) | **< .001** | 1.33 (1.10, 1.60) | **0.004** |
| **Place of residence** | | | | | | |
| | Rural | | Reference | | Reference | |
| | Urban | | 1.05 (0.87, 1.27) | 0.596 | 0.74 (0.61, 0.91) | **0.004** |
| **Intention of pregnancy** | | | | | | |
| | Wanted then | | Reference | | Reference | |
| | Wanted later | | 0.58 (0.51, 0.67) | **< .001** | 0.60 (0.52, 0.69) | **< .001** |
| | No more | | 0.53 (0.44, 0.65) | **< .001** | 0.67 (0.55, 0.82) | **< .001** |
| **Birth order** | | | | | | |
| | 1st | | Reference | | Reference | |
| | 2nd | | 1.09 (0.92, 1.30) | 0.328 | 1.12 (0.92, 1.37) | 0.251 |
| | 3rd | | 1.06 (0.88, 1.27) | 0.563 | 1.14 (0.91, 1.43) | 0.258 |
| | 4th and above | | 0.76 (0.64, 0.91) | **0.003** | 0.94 (0.74, 1.21) | 0.646 |
| **Distance to health facility** | | | | | | |
| | Not a big problem | | Reference | | Reference | |
| | Big problem | | 0.75 (0.64, 0.88) | **0.003** | 0.82 (0.70, 0.96) | **0.013** |
| **ANC provider** | | | | | | |
| | Doctor | | Reference | | Reference | |
| | Nurse or midwife | | 0.50 (0.37, 0.68) | **< .001** | 0.63 (0.47, 0.85) | **0.003** |
| | Auxiliary nurse | | 0.18 (0.04, 0.75) | **0.019** | 0.19 (0.04, 0.82) | **0.026** |

Abbreviations: OR: Odds Ratio; CI: Confidence Interval

1.33: CI = 1.10, 1.60; p = 0.004), and urban women were less likely to receive adequate ANC as compared to rural women (AOR 0.74: CI = 0.61, 0.91; p = 0.004). Compared to women who wanted the current pregnancy, the odds of having adequate ANC were significantly lower for women who wanted a child later (AOR 0.60: CI = 0.52, 0.69; p = 0.001) and those who never wanted a pregnancy (AOR 0.67: CI = 0.55, 0.82; p<0.001). Women who perceived distance to a health facility as a big problem had lower odds of receiving adequate ANC than those who perceived distance to a health facility as not a big problem (AOR 0.82: CI = 0.70, 0.96; p = 0.013). Compared to women whose ANC was provided by doctors, the odds of receiving adequate ANC were lower for women whose ANC was provided by nurses or midwives (AOR 0.63: CI = 0.47, 0.85; p = 0.003), and auxiliary nurses (AOR 0.19: CI = 0.04, 0.82; p = 0.026).

## Discussion

Adequate ANC services increase the possibility of a good pregnancy outcome, yet only 27.62% of women in this secondary analysis of the most recent RDHS received adequate care. It appears that nearly three-quarters of the women faced a challenge accessing ANC in the first three months of pregnancy, attending four visits, and receiving four of the identified ANC components. Studies from high-income countries also disclosed issues related to inadequate ANC and its association with adverse pregnancy outcomes in Canada [14] and France [15]. The high proportion of inadequate ANC in many countries contrasts with WHO recommendations [6].

Findings from the bivariate analysis between adequate ANC receipt and sociodemographic factors showed similarities with other studies. Women were more likely to receive adequate ANC when older (25 to 34 years) than younger women [16]. This finding likely signifies that as the mother's age increases, so does her knowledge, understanding, and experiences of pregnancy and related complications [17, 18]. The benefit of secondary or higher education rather than lower education [19–21] could relate to increased awareness of health services, pregnancy complications, and health-seeking behavior [19]. Being married would likely render partner support and awareness of the ANC benefit to both the mother and child, whereas single mothers would likely lack support [13, 21, 22]. Belonging to the rich and middle-class wealth index rather than the poor index was related to receiving adequate ANC [4, 23–25], with the latter being less likely to have health insurance and delayed access to ANC [26–28].

Other findings from the bivariate analysis between adequate ANC and obstetrical history, and distance to the ANC site revealed further similarities with other studies. Women with unplanned pregnancies were significantly associated with inadequate ANC [29–31], indicating women may keep quiet in fear of frustration or facing stigma with yet another pregnancy. Higher birth order was negatively associated with receiving adequate ANC [31–33] and this finding is regrettable considering that multiparous women are at higher risk of pregnancy and childbirth-related complications such as hypertensive disorders, placenta praevia, and postpartum hemorrhage [32]. Women who felt it was a big problem to reach the health facility were more likely to receive inadequate ANC services [33, 34], possibly due to higher costs and travel time to the health center, especially for mothers with other children.

In Rwanda, most ANC services are provided by nurses and midwives (92.9%) at the primary healthcare level, namely health centers and health posts [12]. The findings revealed that the type of ANC health care providers were significantly associated with women receiving inadequate ANC, and this could be related to the higher acuity attending these primary care services and staff shortages in the rural areas, as in Uganda [34]. In addition, women may receive a delayed referral of serious cases or conflicting information while attending rural ANC services, as in two studies in Rwanda [35, 36]. Other factors exacerbating this situation

may include a lack of equipment and supplies, as reported in a recent systematic review by Dahab et al. on the barriers to accessing maternal care in eight African low-income countries [37]. Our findings reveal that more work needs to be done to assist nurses and midwives in meeting their patients' needs and for the country to meet the Sustainable Development Goals (SDG) by 2030.

In the multivariate logistic regression analysis, variables such as age, marital status, education, and birth order lose their significance compared to bivariate analysis. Being employed or unemployed was not significantly associated with receiving adequate ANC. Conversely, being employed did increase the odds of early initiation of ANC according to a SSA systematic review by Okedo–Alex et al. [26]. Surprisingly, the place of residence was not significant in the bivariate model, but became statistically associated with adequate ANC receipt in the multivariate logistic model, similar to a DHS by Tessema et al. and a multi-country study in SSA by Tekelab et al. [38, 39]. Women living in urban areas may receive adequate ANC due to having many opportunities to access health facilities, in addition to higher educational levels and more knowledge of pregnancy and childbirth risks and danger signs [40].

Since this study was a secondary analysis of a national survey, it would be valuable to explore a deeper understanding of the women's perceptions and beliefs about adherence to adequate ANC through interviews or focus groups. It would also be interesting to research if a first-trimester ultrasound would make a difference to pregnant women coming in earlier for an ANC visit. In high-income countries, prenatal screening in the first trimester is performed for women of advanced maternal age to assess the fetus for Down's syndrome and trisomy 13, 18, and 21 [41]. Furthermore, a study is needed on the HCPs' perspectives of ANC standards, as well as the barriers and enablers to providing the WHO 2016 recommended ANC services.

## Strengths and limitations of the study

The study represented women of childbearing age with an adequate sample size that could be generalized to similar settings. This secondary analysis was conducted using data from responses that were collected within the last three years, and therefore findings from this study helps in identifying the factors associated with ANC. Accordingly, it may help health care system to identify the interventions and implement them to improve ANC in the study area. As with other secondary analyses and cross-sectional surveys, the findings only show a snapshot of a person one time; there is a potential for recall bias, and we could not explore other factors related to adequate ANC found in primary studies.

## Conclusion

The current study showed that only about a quarter of the pregnant women surveyed in the most recent DHS received adequate ANC in Rwanda. The women's wealth index, health insurance, residence location, pregnancy intention, distance to a health center, and ANC provider were associated with the receipt of adequate ANC. There is an urgent need for interventions to increase ANC utilization in Rwanda to meet the SDG, starting with the factors identified in this study.

## Acknowledgments

We greatly acknowledge the Measure DHS program granting access to the 2019–20 Rwanda Demographic Health Survey data sets.

## Author Contributions

**Conceptualization:** Olive Tengera, Laetitia Nyirazinyoye, Pamela Meharry, Stephen Rulisa, Zelalem T. Haile.

**Data curation:** Olive Tengera, Laetitia Nyirazinyoye, Pamela Meharry, Reverien Rutayisire, Zelalem T. Haile.

**Formal analysis:** Olive Tengera, Laetitia Nyirazinyoye, Pamela Meharry, Reverien Rutayisire, Zelalem T. Haile.

**Investigation:** Olive Tengera.

**Methodology:** Olive Tengera, Laetitia Nyirazinyoye, Pamela Meharry, Zelalem T. Haile.

**Project administration:** Olive Tengera.

**Visualization:** Olive Tengera, Laetitia Nyirazinyoye, Pamela Meharry, Stephen Rulisa, Zelalem T. Haile.

**Writing – original draft:** Olive Tengera, Laetitia Nyirazinyoye, Pamela Meharry, Stephen Rulisa, Zelalem T. Haile.

**Writing – review & editing:** Olive Tengera, Laetitia Nyirazinyoye, Pamela Meharry, Reverien Rutayisire, Stephen Rulisa, Zelalem T. Haile.

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
