## [Decision Letter · Decision Letter 0]

11 Aug 2022

PONE-D-22-19638Factors associated with receipt of adequate antenatal care during pregnancy among women in Rwanda: A population-based cross-sectional studyPLOS ONE

Dear Author,

Thank you for submitting your manuscript to PLOS ONE. After careful consideration, we feel that it has merit but does not fully meet PLOS ONE’s publication criteria as it currently stands. Therefore, we invite you to submit a revised version of the manuscript that addresses the points raised during the review process. ==============================

ACADEMIC EDITOR: Please format the manuscript according to the journal requirements https://journals.plos.org/plosone/s/submission-guidelines.Please refer to PLOS ONE downloadable sample files to ensure that your submission meets the journal formatting guidelinesPlease ensure you have followed STROBE guidelines for cross-sectional studies Pease make sure that ALL abbreviations are explained in the main text. Line 55 in the introduction - "ANC" abbreviation is not defined.==============================

We look forward to receiving your revised manuscript.

Kind regards,

Gulzhanat Aimagambetova

Academic Editor

PLOS ONE

Journal Requirements:

a) Did participants provide their written or verbal informed consent to participate in this study?

Reviewers' comments:

Reviewer's Responses to Questions

**Comments to the Author**

1. Is the manuscript technically sound, and do the data support the conclusions?

Reviewer #1: Partly

Reviewer #2: Yes

2. Has the statistical analysis been performed appropriately and rigorously? 

Reviewer #1: No

Reviewer #2: Yes

3. Have the authors made all data underlying the findings in their manuscript fully available?

Reviewer #1: No

Reviewer #2: Yes

4. Is the manuscript presented in an intelligible fashion and written in standard English?

Reviewer #1: Yes

Reviewer #2: No

5. Review Comments to the Author

Reviewer #1: Authors in the current study made great efforts to assess the receipt of adequate ANC among pregnant women in the study area and its associated factors. However, I do have some concerns over the methods, results and discussion part which I think need to be addressed before this can be published. Although I am not a statistician, I tried to think in a logical manner as far as results are concerned. So, I apologies, if my lack of expertise in this field has led to confusion. My comments are as follows:

Major comments:

1. Regarding the study design, I understood (line no.99) that the current study used the data from RDHS(a cross sectional survey). If it is so, the current study cannot be a cross-sectional study, I feel. The study used part of data from 2019-20 RDHS, therefore, it could be a secondary analysis, as observed in a similar study conducted and published in 2016 by Rwabufigir BN et al. (ref 18 of the current version of the manuscript) where they clearly mentioned about the study design. Please relook into their study design and rewrite accordingly. If my understanding is not correct, please clarify.

2. Under study variables, the manuscript discussed that no. of visits was coded (line no. 121). But, the manuscript failed to interpret these findings. I am trying to think louder on this particular variable. Somehow, this variable may allow the policy makers to think about the bottlenecks in the delivery of adequate ANC services, thereby, they can plan some interventions like providing resources, IEC or awareness etc. For instance, even if X no of pregnant women completed atleast four visits of ANC, whether all of these X subjects received adequate ANC (in terms of others components-time of first visit and receipt of the four components).

I assume that authors included all these three components under one variable named ‘Adequate ANC’. However, results can be more appreciated if the manuscript provides these estimates as well. If you disagree of this point, please justify. If agree, kindly present all the missing data.

3. Under the statistical analysis of methods section, kindly elaborate the variables (line no. 145) and factors that are considered in the bivariate analysis. Like demographics, socio economic characters etc.

4. Results section:

a. Kindly rewrite the title of Table 1. It would be better to mention this descriptive statistics in the methods sections rather than in the title. As these are demographics and socio-economic characters of study population, kindly relook into it.

b. To bring the uniformity in the age groups and make them comparable, it is advisable to split the last group “35-49 years” into two groups as ‘35-44 years’ and ’45 years and above’. If agreed, the further calculations in the Table 2 and Table 3 will also be varied accordingly.

c. When I go through the results, there is a slight confusion in the number of participants considered in the multivariate analysis. Throughout the manuscript, the study aimed at identifying the factor associated with the adequate ANC services. In the Table 1, it was already estimated that 27.6% of study population received adequate ANC. If author wants to represent the demographic and socio-economic characters of adequate ANC received population, the sample size cannot be 6039 (n) i.e., try to avoid the inadequate ANC population. If author wishes to represent characters of all (n=6039) participants, the title of Table 2 should not include the word ‘adequate ANC’. Indeed, my suggestion would be to merge the Table 1 and 2 and change the results interpretation accordingly. Kindly address.

d. The current study aimed to estimate the factors associated with adequate ANC, but not the ANC utilization in the study area, if I am not wrong. Therefore, total population (n=6039) cannot be used in multivariate analysis (provided in Table 3), when the Adequate ANC was found to be on 27.6% (n=1691). Accordingly, results, discussion and conclusion of the study may vary. I strongly suggest taking recommendation/expert opinion from Biostatistician on this matter.

e. It is advisable to categorize birth order variable as 1, 2, 3 and 4 and above. Accordingly find the association.

5. In the discussion section, from 204-206, there is a problem in the references quoted. 18th reference is on post natal care. In line no. 206, I think, 17th reference to be quoted instead of 18th reference in the current version of manuscript. Kindly relook into these references (i.e.17, 18, &19) and address properly. Also kindly make necessary changes in the order of references as appropriate.

Minor comments:

1. Since the current study has not performed the correlation tests, the term co-variates in line no. 115, 129 may be replaced with variables.

2. In Table 2, kindly relook into the percentages provided under the following variable-

a. Age group & Education (calculation is not correct)

b. Health Insurance & Distance (please make it to 100%)

3. In line no. 204, references (supposed to be 17,18) are in wrong order.

4. Suggestion to change the title: since antenatal care provides during pregnancy, slight corrections may be made in the title like Factors associated with adequate antenatal care among pregnant women in Rwanda......... If this study is not a primary study conducted by the authors, population based cross-sectional study can be replaced with better terms to make it more appropriate.

5. References: strictly adhere to the journal guidelines.

Reviewer #2: topic is interesting enough to attract the readers’ attention. Nevertheless, authors should clarify some points and improve the discussion, as suggested below.

Authors should consider the following recommendations:

- Manuscript should be further revised in order to correct some typos and improve style.

- Authors should ad further details to discuss, at least briefly, the role of first trimester screening for fetal aneuploidies, especially in advanced maternal age (authors may refer to: PMID: 27442264; PMID: 25027820).

6. PLOS authors have the option to publish the peer review history of their article (what does this mean?). If published, this will include your full peer review and any attached files.

Reviewer #1: **Yes: **Dr Kayala Venkata Jagadeesh

Reviewer #2: No

---

## [Author Response · Author response to Decision Letter 0]

17 Jan 2023

All given comments were addressed

---

## [Decision Letter · Decision Letter 1]

24 Feb 2023

PONE-D-22-19638R1Factors associated with receipt of adequate antenatal care among women in Rwanda: A secondary analysis of 2019-20 Demographic and health survey dataPLOS ONE

Dear Dr. Olive Tengera,

Thank you for submitting your manuscript to PLOS ONE. After careful consideration, we feel that it has merit but does not fully meet PLOS ONE’s publication criteria as it currently stands. Therefore, we invite you to submit a revised version of the manuscript that addresses the points raised during the review process.

We look forward to receiving your revised manuscript.

Kind regards,

Gulzhanat Aimagambetova

Academic Editor

PLOS ONE

Journal Requirements:

Reviewers' comments:

Reviewer's Responses to Questions

**Comments to the Author**

1. If the authors have adequately addressed your comments raised in a previous round of review and you feel that this manuscript is now acceptable for publication, you may indicate that here to bypass the “Comments to the Author” section, enter your conflict of interest statement in the “Confidential to Editor” section, and submit your "Accept" recommendation.

Reviewer #1: All comments have been addressed

Reviewer #2: All comments have been addressed

2. Is the manuscript technically sound, and do the data support the conclusions?

Reviewer #1: Yes

Reviewer #2: Yes

3. Has the statistical analysis been performed appropriately and rigorously? 

Reviewer #1: Yes

Reviewer #2: Yes

4. Have the authors made all data underlying the findings in their manuscript fully available?

Reviewer #1: Yes

Reviewer #2: Yes

5. Is the manuscript presented in an intelligible fashion and written in standard English?

Reviewer #1: Yes

Reviewer #2: Yes

6. Review Comments to the Author

Reviewer #1: Thank you for addressing the comments. Kindly look into following observations made:

1. In the methodology, You mentioned about Chi square test under statistical analysis. However, no where it is mentioned in the results section. Have authors only considered for estimating p-value? Kindly justify.

2. Kindly re-looks into the interpretation of results at line 184-186. This result may be interpreted other way round.

3. In the Table-3 title, can we not use 'ANC utilization' instead of 'ANC Status'. Because the interpretation of results given in table-3 are all about receiving ANC. Kindly relook into it.

4. References were not written properly. Please strictly adhere to the journal author guidelines. For example, ref 11 is not written completely. There are other issues in rest of the reference. Kindly cross check all the reference again.

Reviewer #2: I carefully evaluated the revised version of this manuscript.

Authors have performed the required changes, improving significantly the quality of the paper.

7. PLOS authors have the option to publish the peer review history of their article (what does this mean?). If published, this will include your full peer review and any attached files.

Reviewer #1: **Yes: **Dr Kayala V Jagadeesh

Reviewer #2: No

---

## [Decision Letter · Decision Letter 2]

20 Mar 2023

PONE-D-22-19638R2Factors associated with receipt of adequate antenatal care among women in Rwanda: A secondary analysis of 2019-20 Rwanda Demographic and Health SurveyPLOS ONE

Dear Dr. Tengera,

Thank you for submitting your manuscript to PLOS ONE. After careful consideration, we feel that it has merit but does not fully meet PLOS ONE’s publication criteria as it currently stands. Therefore, we invite you to submit a revised version of the manuscript that addresses the points raised during the review process.

We look forward to receiving your revised manuscript.

Kind regards,

Gulzhanat Aimagambetova

Academic Editor

PLOS ONE

Journal Requirements:

Reviewers' comments:

Reviewer's Responses to Questions

**Comments to the Author**

1. If the authors have adequately addressed your comments raised in a previous round of review and you feel that this manuscript is now acceptable for publication, you may indicate that here to bypass the “Comments to the Author” section, enter your conflict of interest statement in the “Confidential to Editor” section, and submit your "Accept" recommendation.

Reviewer #1: All comments have been addressed

2. Is the manuscript technically sound, and do the data support the conclusions?

Reviewer #1: Yes

3. Has the statistical analysis been performed appropriately and rigorously? 

Reviewer #1: Yes

4. Have the authors made all data underlying the findings in their manuscript fully available?

Reviewer #1: Yes

5. Is the manuscript presented in an intelligible fashion and written in standard English?

Reviewer #1: (No Response)

6. Review Comments to the Author

**Reviewer #1**: Thank you for addressing the comments. However, I have following observations in the manuscript:

1. In the Methodology, at line 138 under Ethics approval, a minor language correction is required. Kindly, you may use ‘which was’ instead of ‘and’ (before registered).

2. In title of table-2, does the sub-heading ‘adequacy ANC received’ or ‘Adequacy of ANC received’? Please use appropriate sub-heading.

3. A suggestion from my side about a point that starts at Line 181 till 184 as follows –

‘Women from a middle and rich wealth indexes had higher odds of receiving adequate ANC ((AOR=1.24: CI=1.04-1.48; p=0.018) and (AOR=1.37: CI=1.16-1.61; p<0.001) respectively) compared to women from poor household wealth index’

4. Point starts at end of line no. 189 till 191 is misinterpreted, as per the results provided in Table-3 regarding distance to health facility. Kindly re-look into it.

5. At line 261, current study is not directly providing recommendations on interventions. Perhaps, it helps in identifying the factors associated with ANC. Accordingly, it may help health care system to identify the interventions and implement them to improve ANC in the study area. If you agree, kindly re-write the statement (at end of line 261 and 262)

6. References: Total references quoted in the text are 40, whereas under references sections, authors have listed 41. Please check and correct it.

7. Authors are also requested to adhere to Journal author guidelines while writing the reference. In most of the references, month was written. For example, look at ref. no. 4, 16, 17, 18 and others. Please cross check all the references again.

7. PLOS authors have the option to publish the peer review history of their article (what does this mean?). If published, this will include your full peer review and any attached files.

Reviewer #1: No

---

## [Decision Letter · Decision Letter 3]

6 Apr 2023

Factors associated with receipt of adequate antenatal care among women in Rwanda: A secondary analysis of 2019-20 Rwanda Demographic and Health Survey

PONE-D-22-19638R3

Dear Dr. Olive Tengera

We’re pleased to inform you that your manuscript has been judged scientifically suitable for publication and will be formally accepted for publication once it meets all outstanding technical requirements.

Kind regards,

Gulzhanat Aimagambetova

Academic Editor

PLOS ONE

Reviewers' comments:

Reviewer's Responses to Questions

**Comments to the Author**

1. If the authors have adequately addressed your comments raised in a previous round of review and you feel that this manuscript is now acceptable for publication, you may indicate that here to bypass the “Comments to the Author” section, enter your conflict of interest statement in the “Confidential to Editor” section, and submit your "Accept" recommendation.

Reviewer #1: All comments have been addressed

2. Is the manuscript technically sound, and do the data support the conclusions?

Reviewer #1: Yes

3. Has the statistical analysis been performed appropriately and rigorously? 

Reviewer #1: Yes

4. Have the authors made all data underlying the findings in their manuscript fully available?

Reviewer #1: Yes

5. Is the manuscript presented in an intelligible fashion and written in standard English?

Reviewer #1: Yes

6. Review Comments to the Author

Reviewer #1: No comments. Please check the references again, as page numbers are either missing or written incorrectly/typo in some references provided.

7. PLOS authors have the option to publish the peer review history of their article (what does this mean?). If published, this will include your full peer review and any attached files.

Reviewer #1: **Yes: **Kayala Venkata Jagadeesh

---

## [Editor Report · Acceptance letter]

12 Apr 2023

PONE-D-22-19638R3 

Factors associated with receipt of adequate antenatal care among women in Rwanda: A secondary analysis of the 2019-20 Rwanda Demographic and Health Survey 

Dear Dr. Tengera:

I'm pleased to inform you that your manuscript has been deemed suitable for publication in PLOS ONE. Congratulations! Your manuscript is now with our production department. 

Kind regards, 

on behalf of

Dr. Gulzhanat Aimagambetova 

Academic Editor

PLOS ONE